# Mucopolysaccharidosis-Plus Syndrome: Is This a Type of Mucopolysaccharidosis or a Separate Kind of Metabolic Disease?

**DOI:** 10.3390/ijms25179570

**Published:** 2024-09-04

**Authors:** Zuzanna Cyske, Lidia Gaffke, Karolina Pierzynowska, Grzegorz Węgrzyn

**Affiliations:** Department of Molecular Biology, Faculty of Biology, University of Gdansk, Wita Stwosza 59, 80-308 Gdansk, Poland; zuzanna.cyske@ug.edu.pl (Z.C.); lidia.gaffke@ug.edu.pl (L.G.); karolina.pierzynowska@ug.edu.pl (K.P.)

**Keywords:** mucopolysaccharidosis-plus syndrome, mucopolysaccharidosis, glycosaminoglycans, vesicular trafficking

## Abstract

Several years ago, dozens of cases were described in patients with symptoms very similar to mucopolysaccharidosis (MPS). This new disease entity was described as mucopolysaccharidosis-plus syndrome (MPSPS). The name of the disease indicates that in addition to the typical symptoms of conventional MPS, patients develop other features such as congenital heart defects and kidney and hematopoietic system disorders. The symptoms are highly advanced, and patients usually do not survive past the second year of life. MPSPS is inherited in an autosomal recessive manner and is caused by a homozygous-specific mutation in the gene encoding the VPS33A protein. To date, it has been described in 41 patients. Patients with MPSPS exhibited excessive excretion of glycosaminoglycans (GAGs) in the urine and exceptionally high levels of heparan sulfate in the plasma, but the accumulation of substrates is not caused by a decrease in the activity of any lysosomal enzymes. Here, we discuss the pathomechanisms and symptoms of MPSPS, comparing them to those of MPS. Moreover, we asked the question whether MPSPS should be classified as a type of MPS or a separate disease, as contrary to ‘classical’ MPS types, despite GAG accumulation, no defects in lysosomal enzymes responsible for degradation of these compounds could be detected in MPSPS. The molecular mechanism of the appearance of GAG accumulation in MPSPS is suggested on the basis of results available in the literature.

## 1. Introduction—Mucopolysaccharidosis

Mucopolysaccharidosis (MPS), belonging to the group of lysosomal storage diseases (LSD), is a hereditary, progressive disease caused by the complete lack or low residual activity of lysosomal enzymes responsible for the degradation of unbranched sugar chains—glycosaminoglycans (GAGs). The undegraded GAGs gradually accumulate in the lysosomes, slowly impairing their functions [1]. The activity of the aforementioned enzymes is closely interconnected, with each enzyme starting its function only after the previous one has completed its catalytical reaction. Therefore, in the absence of the activity of even one enzyme, the entire pathway ceases to function correctly. There are 13 types/subtypes of MPS, classified based on the inactive or highly deficient enzyme and the accumulating GAG(s) [2].

MPS is inherited in an autosomal recessive manner except for MPS II (Hunter syndrome), which is X-linked. Depending on the type, the disease occurs with varying frequency. MPS I is the most common variant, occurring in about 1 in 88,000 live births, while MPS VII occurs in 1 in 2,111,000 live births. MPS IX is the rarest type, with only four cases described worldwide to date [3]. There are many symptoms characteristic of all patients with MPS, but there are also symptoms typical for specific types/subtypes. Although the mutations leading to the development of various types of MPS involve different genes, they result in one outcome: the accumulation of GAG(s) in lysosomes [4]. The most commonly used method for the preliminary diagnosis of MPS is the measurement of GAGs in a urine sample. Subsequent stages of the diagnostic process include determining the activity of lysosomal enzymes and then searching for specific mutations in the identified gene [5].

For many years, the assistance for patients affected by MPS was limited to palliative care. A possible therapy turned out to be a bone marrow transplantation [6]. However, this therapy is only effective if the disease is diagnosed very quickly, that is, before the first symptoms appear. It is not possible to reverse existing neurological defects, such as cognitive functions, hyperactivity, and emotional disturbances. The initiation of the work on enzyme replacement therapy (ERT), which involves the intravenous administration of the active form of the deficient enzyme, led to a departure from the routine use of bone marrow transplants. The benefits of ERT for patients with MPS include improved joint mobility, and consequently, improved mobility; improved respiratory function, reduced liver size, and a significant decrease in the amount of GAG(s) excreted in the urine [7,8,9,10]. However, ERT cannot eliminate all symptoms affecting patients with MPS, and most importantly, the enzyme used in the therapy cannot cross the blood–brain barrier, making the treatment ineffective for types with central nervous system symptoms. Due to the need to treat neuronopathic types of MPS, an alternative form of therapy has been proposed: substrate reduction therapy (SRT) [4]. This therapy focuses on trying to reduce the efficiency of GAG synthesis (given the reduced efficiency of their degradation), which restores the balance between their production and decay.

## 2. Mucopolysaccharidosis-Plus Syndrome—General Features

At the beginning of 2010, an LSD leading to death before the age of two years was described in the Republic of Sakha (Yakutia, Russia). It was diagnosed as an ‘undifferentiated hereditary metabolic disease’ [11]. In 2017, after the clinical picture was described and the cause of the disease was identified, the new disease was named mucopolysaccharidosis-plus syndrome (MPSPS). The name of the disease indicates that, in addition to the typical symptoms of conventional MPS, patients developed other features such as congenital heart defects, kidney dysfunctions, and hematopoietic system disorders. MPSPS is inherited in an autosomal recessive manner and was initially reported to be caused by a homozygous-specific mutation (c.1492C>T; p.Arg498Trp) in the gene encoding the VPS33A protein. To date, it has been described in 39 out of 41 patients reported in the literature. It is estimated that the incidence of MPSPS in the Yakut population is 1 in 12,100 births [12,13], but it is much rarer in other populations. Patients with MPSPS exhibited excessive excretion of GAGs in the urine and exceptionally high levels of heparan sulfate in the plasma. These results indicated that dysfunction of the VPS33A protein leads to GAG accumulation and causes a phenotype similar to MPS. However, the accumulation of substrates is not caused by a decrease in the activity of any known lysosomal enzymes.

Recently, the first two patients with MPSPS outside Yakutia and Turkey were diagnosed. These patients exhibited symptoms similar to, but somewhat milder than, those previously described in MPSPS patients. One of them was a young adult of Southern European/Mediterranean origin [14], while the second was a 12-year-old patient from Poland [15]. It is clear that the life spans of these two patients were significantly longer than those of previously described patients with the c.1492C>T (p.Arg498Trp) mutation. In both cases of the juvenile MPSPS patients, genetic analyses (whole-exome sequencing) indicated the presence of a homozygous pathogenic variant of the *VPS33A* gene (c.599G>C; p.Arg200Pro). Apart from symptoms similar to those described previously in other MPSPS patients, the Polish patient revealed some additional features, like recurrent joint effusion, peripheral edemas, and visceral obesity, while being of normal growth. Therefore, one might suggest that the c.599G>C variant may be responsible for the milder phenotype than that caused by the c.1492C>T mutation.

## 3. Excretion of Glycosaminoglycans and Mutations in the *VPS33A* Gene

As in all types/subtypes of MPS, patients suffering from MPSPS also exhibit increased excretion of GAGs, mainly an excess of heparan sulfate, dermatan sulfate, and chondroitin sulfate in urine and plasma [13,14,15,16]. In one patient, the presence of keratan sulfate was also detected [16]. Studies have also shown increased amounts of sialooligosaccharides and sialic acid in the urine of patients with MPSPS [15,16,17].

The *VPS33A* gene is located on chromosome 12q24.31 and contains 13 exons. The product of this gene, the VPS33A protein, consists of 596 amino acid residues and has a molecular weight of approximately 67 kDa. It is composed of four domains: 1, 2, 3a, and 3b [18,19]. Different domains within the VPS33A protein are responsible for its various functions. It is predicted that domain 3a may interact with the SNARE complex [18,20]. Domain 3b binds to the VPS16 protein (a subunit of the HOPS complex) [19]. The c.1492C>T (p.Arg498Trp) mutation in the *VPS33A* gene, described in most MPSPS patients, is located in domain 2 of the VPS33A protein. Currently, domains 1 and 2 of VPS33A are those of unknown functions [11]. However, it was demonstrated that the c.1492C>T mutation in the *VPS33A* gene affects GAG metabolism [13]. Arginine residue at position 498 is highly conserved across many species. In silico analysis indicated that the p.Arg498Trp mutation is disease-causing and decreases the stability of the protein.

Patients with MPSPS showed excessive GAG excretion in urine and high levels of heparan sulfate in plasma. These results suggested that reduced VPS33A protein activity may lead to GAG accumulation and cause phenotypes similar to MPS I (Hurler syndrome). However, the accumulation of GAGs was not due to the reduced activity of any known lysosomal enzymes involved in GAG degradation. The mutation also did not affect the localization of these enzymes and their substrates (GAGs), cathepsin D processing, or lipid transport. The c.1492C>T mutation did not affect endocytic and autophagic pathways, as examination of autophagy revealed no disturbances in this process in fibroblasts derived from patients. A study of the binding of the VPS33A mutant variant (p.Arg498Trp) to its known partners, VPS16 (HOPS and CORVET) and STX17, showed that these interactions were unchanged. In cells derived from patients, slight excessive acidification of lysosomes was observed [11]. On the other hand, the conclusion about unaffected intracellular trafficking in MPSPS cells came from only a couple of experiments, namely, colocalization of heparan sulfate with lysosomal enzymes and membranes, and on normal levels of filipin [13].

Only two MPSPS patients have been described with a mutation ((c.599G>C; p.Arg200Pro) in the *VPS33A* gene) other than the previously known (c.1492C>T) and considered for some time as the only one occurring in MPSPS patients [14,15]. As mentioned above, the c.599G>C mutation presents with a milder phenotype than that observed in other patients, and the affected individuals live significantly longer than those with the c.1492C>T mutation [14,15]. Molecular analyses suggested that the deficiency of the enzymatic activity of the product of the c.599G>C variant of *VPS33A* mostly results from decreased stability of the protein, which is more prone to proteasomal degradation than the wild-type variant [14].

## 4. Why Glycosaminoglycans Accumulate in MPSPS—A Hypothesis

Mucopolysaccharidoses (MPS) are diseases caused by mutations in genes encoding lysosomal enzymes involved in the degradation of GAGs. Recent studies indicated that changes in the expression of many genes may cause secondary and tertiary cellular dysfunctions, influencing the course of these diseases. The changes included sequences coding for both proteins and regulatory RNAs [21,22,23]. It was shown that apart from other changes, the regulation of vesicle trafficking is impaired in cells from patients suffering from various types of MPS due to changes in the levels of key proteins involved in this process [24]. Therefore, one might ask whether vesicular transport of GAGs might be impaired in MPSPS, leading to their improper localization and thus inefficient degradation.

We assume that although colocalization of GAGs with lysosomal enzymes in MPSPS cells was reported [13], the colocalization does not necessarily determine the intra-lysosomal localization of GAGs, which could also be present in the cytoplasm. Moreover, normal levels of filipin (demonstrated previously in MPSPS [13]) suggested unaffected lipid trafficking indeed; however, specific GAGs can be transported through different pathways, dependent on either caveolin or clathrin [25,26]. Therefore, as suggested previously [24], the potential defective transport of GAGs in MPSPS cells might lead to their accumulation. Such a proposal might be corroborated by the already demonstrated impairment of intracellular glycosphingolipid trafficking. Taking this into consideration, our hypothesis is that the mutation in the *VPS33A* gene, through resulting in impaired VPS33A protein function or stability, impairs vesicular transport so that GAGs cannot be properly transported to lysosomes and degraded. The scheme of the putative MPSPS pathomechanism, based on the accumulation of GAGs, is presented in Figure 1.

As mentioned above, an excess of different GAGs, especially heparan sulfate, dermatan sulfate, and chondroitin sulfate, occurs in the urine and plasma of MPSPS patients [13,14,15,16]. Since all GAGs are degraded by lysosomal hydrolases, significant differences in the defects of intra-lysosomal decay of specific GAGs in MPSPS are unlikely. Thus, we assume that the accumulation of specific, above-mentioned GAGs might be related to the mechanisms of their transport to lysosomes. Indeed, there are different pathways of transportation of GAGs into lysosomes. For example, heparan sulfate and hyaluronate are transported inside the cell by employing caveolae-dependent endocytosis, while dermatan sulfate might also be transported by the clathrin-promoted process [25,26]. Importantly, recent studies demonstrated that caveolin is significantly less abundant in MPS cells than in normal ones, apparently due to impaired *CAV1* gene expression [24]. In MPS, the primary GAG storage (arising from dysfunctional lysosomal hydrolases) causes stress conditions, which result in (among others) disrupted cell signaling and dysregulation of the expression of many genes. The dysregulated genes include those encoding caveolin and other proteins involved in caveolin-dependent endocytosis and vesicle trafficking [24]. In the case of MPS, this may lead to the accumulation of undegraded GAGs not only in lysosomes but also in the cytoplasm and outside the cells. If so, impaired transport of GAGs into lysosomes further stimulates pathological processes due to even more pronounced cellular stresses and changes in gene expression regulation. In the case of MPSPS, the disturbed GAG transport (caused by VPS33A dysfunction) could be the initial, rather than a secondary step of the spiral of the above-described events; nevertheless, causing positive feedback leading to more and more efficient GAG accumulation (Figure 1).

An alternative hypothesis is that impaired GAG degradation in MPSPS is due to increased endolysosomal acidification, demonstrated previously in patient-derived cells [13,14]. However, if this hypothesis were true, one should observe the accumulation of not only GAGs but also many other compounds, like proteins, lipids, and others, due to general dysfunctions of lysosomes, similar to the processes occurring in mucolipidosis (ML) type II and III, where lysosomal enzymes are improperly localized, thus being unable to catalyze specific reactions. Contrary to ML II and III, such defects in the degradation of very different macromolecules were not reported in MPSPS.

## 5. Symptoms of MPSPS in Comparison to Those Occurring in Classical MPS

All children suffering from the infant form of MPSPS developed extremely severe symptoms leading to death before the age of two years. A few years after the disease was described, it was included in the group of mucopolysaccharidoses, with the addition of ‘plus’. This term refers to the fact that, in addition to the symptoms present in all other types/subtypes of MPS, additional symptoms such as kidney dysfunction, hematopoietic system disorders, and congenital heart defects also developed.

Children with MPSPS exhibit characteristic facial features, facial and limb swelling, and loose skin, similar to other types/subtypes of MPS [13,14,15,17,27]. Additionally, severe enlargement and dysfunction of the liver and spleen have been observed in both types/subtypes of MPS and MPSPS [11,13,14,15,27].

Hematopoietic system disorders are also present in MPSPS. Most patients exhibit a decrease in red blood cells and hemoglobin levels [28], as well as normocytic anemia, thrombocytopenia, neutropenia, and coagulation disorders [16,17]. The hypogammaglobulinemia seen in patients may be responsible for impaired humoral immunity and frequent recurrent infections [16]. The autopsy results of patients showed the presence of hypoplastic bone marrow [13,17].

### 5.1. Symptoms Detectable during Prenatal Testing

Prenatal testing is extremely important for the diagnosis of genetic diseases and the selection of appropriate treatments. Early diagnosis allows for immediate treatment, which can significantly improve the quality of life for affected individuals and may even prevent the onset of symptoms. Prenatal testing for MPS is possible, but due to the rarity of the disease, prospective parents typically do not undergo such tests without a specific reason, such as having an affected first child or knowing they are carriers.

In a patient with MPSPS, prenatal ascites were detected via ultrasound at 11 and confirmed at 24 weeks of pregnancy [15]. A similar symptom was described in a patient with MPS VI, where prenatal symptoms began with ascites and rapidly progressed to mitral valve regurgitation and congestive heart failure, ultimately requiring surgical intervention within the first year of life [29]. Interestingly, all patients with MPSPS have congenital heart defects, such as mitral or aortic valve regurgitation, thickening of the interventricular septum, or atrial enlargement [11,14,17].

Prenatal abnormalities, including edema, fluid in the abdominal cavity, pleural effusion, and polyhydramnios, have also been diagnosed in patients with MPS VII [30]. Patients with MPS VII often suffer from severe prenatal conditions such as NIHF (non-immune hydrops fetalis). About half of patients with MPS VII die due to complications related to NIHF [31]. Similar edema symptoms have been prenatally diagnosed in patients with MPS II. A case was described where siblings were affected; the older child showed left hydronephrosis prenatally and inguinal hernia a month after birth, while the younger child underwent prenatal testing, revealing the presence of a mutation leading to MPS II development [32].

Further studies of women at risk of having children with MPSPS showed increased nasal mucosa thickness in the fetus. In another woman, an ultrasound examination of the child showed an increased nuchal fold and prenasal thickness [27].

### 5.2. Development

In patients with MPSPS, developmental disorders are quickly noticeable compared to healthy children. Patients with MPSPS are typically born on time, but the first symptoms of the disease appear within a few months. Most patients exhibit delayed psychomotor development with autistic symptoms [14,15]. There are also delays in learning to sit and walk: independent sitting after 13 months of age, walking after 28 months of age [15], sitting from 10 months of age, walking with assistance after 22 months of age, and babbling started at 11 months [17]. Usually, at the age of 2 years, patients can sit with support but cannot stand and speak [11]. The motor skills of patients with MPSPS are often limited due to joint stiffness and clawed fingers, resulting in frequent falls and difficulties in holding objects. Concentration and attention difficulties, poor memory, and limited cognitive abilities are also commonly described [14,17].

Similar symptoms are observed in patients with various types/subtypes of MPS. For example, patients with MPS VII show motor development issues, which is evident in unstable walking and frequent falls [30]. A typical patient with MPS VII could sit independently at seven months and begin walking at fourteen months [30]. At the age of 5 years, such a patient still has walking difficulties, a limp, and abnormal posture [30]. Patients with MPS III (Sanfilippo disease) also exhibit similar disorders as those with MPSPS. A child with Sanfilippo A began walking at 2 years, speaking at 3 years, and exhibiting general psychomotor developmental delays while being impulsive and hyperactive [33]. Similar symptoms also occur in patients with other subtypes of MPS III, where, over time, their mental health deteriorates, eventually leading to severe dementia and behavioral problems such as hyperactivity and aggression [34].

Patients with MPSPS often have a shorter stature compared to healthy children of the same age [17]. Similar observations have been made in patients with other types/subtypes of MPS. After birth, their average z-scores for body height were lower compared to reference values [35]. Children with MPS IVA stop growing at the age of 8 years, and their earlier growth is very slow. In MPS VI, growth can be variable during the first four years of life, with abrupt declines. Before the age of 5 years, these changes stabilize and are no longer as abrupt. The average z-scores for other MPS types (MPS I, II, and III) indicate that up to the age of 2 years, patients’ growth is similar, and the average z-scores for body height are higher than in reference charts [36].

### 5.3. Central Nervous System

Patients with MPSPS exhibit severe central nervous system disorders similar to those seen in other types/subtypes of MPS with neurological symptoms (MPS I, II, III, and VII). Neurological examinations show neurodevelopmental delay, and MRI brain scans reveal delayed myelination, brain calcifications, retinal hypopigmentation, cerebellar abnormalities, and, in some cases, global brain atrophy [13,16,17]. However, studies have shown that after treatment initiation and stabilization of the patient’s condition, neurological abilities may gradually improve [17]. Some patients also exhibit nystagmus, which subsides over time in some cases [16,17].

Studies on a mouse model of MPS IIIC revealed similar disorders to those seen in MPSPS, including significantly reduced levels of proteins associated with myelination, such as myelin protein, myelin glycoprotein, and oligodendrocyte myelin glycoprotein. Structural studies showed disorganization of myelin sheaths and loss of myelin thickness in the brains of MPS IIIC animals [37]. In patients with MPS IIID, these disorders were less severe, namely, they exhibited mild cortical and cerebellar atrophy [38]. Patients with MPS I showed similar disorders to those with MPSPS, including severe white matter loss, brain atrophy, and, in some cases, hydrocephalus [39]. MRI studies showed that almost all patients with Sanfilippo disease exhibit cortical atrophy [40].

### 5.4. Heart

Cardiovascular diseases are currently the leading cause of premature death in patients with MPS [41]. Problems with proper diagnosis and a lack of appropriate treatment lead to rapid disease progression and the development of many complications, including heart failure [42,43]. All patients with MPSPS were found to have severe heart function disorders, the most common being mitral valve regurgitation [13,17], as well as aortic and tricuspid valve regurgitation [11]. A patient with a milder mutation (p.Arg200Pro) also showed mitral valve stenosis, but it was significantly milder than in other MPSPS patients and did not progress with age [15]. Additionally, some MPSPS patients were noted to have thickening of the interventricular septum [14] or atrial enlargement [17]. The most common cardiovascular disorder found in patients with other types/subtypes of MPS is mitral valve regurgitation [44]. Systolic murmur, aortic valve stenosis and regurgitation, thickened ends of aortic cusps, and mild left ventricular hypertrophy were observed in MPS X patients. Further studies on the MPS II model confirmed that the most common defect in these patients (56% of those studied) is mitral valve regurgitation, followed by aortic valve defect (33%) [45].

Overall, studies with MPS patients did not show a correlation between patient age and cardiac phenotype. In some cases, a patient with a mild disease course exhibited more severe heart damage than a patient with a severe phenotype [44].

### 5.5. Lungs

Lung diseases are extremely severe and rapidly recognized in patients with MPSPS. Patients are very often hospitalized due to recurrent upper respiratory tract infections and pneumonia [13,17]. In almost every described case, MPSPS patients were admitted to the hospital due to upper respiratory tract infections. A notable example is a female patient hospitalized at 2.5 months, 7 months, and 10 years, each time diagnosed with severe pneumonia [15], as well as a female patient who was repeatedly hospitalized after her first year of life due to respiratory tract infections [11]. Patients also frequently experience episodes of dyspnea and apnea attacks [16].

Patients with other types/subtypes of MPS also very often have recurrent upper and lower respiratory tract infections. Additionally, patients with MPS I frequently exhibit airway obstruction during sleep, which can result in obstructive sleep apnea [46]. Newborns with MPS II have been reported to have a higher incidence of respiratory failure than healthy children [47]. In MPS IV, a reduced chest size compared to healthy children and obstructive lung disease have been noted. Diaphragm displacement associated with hepatosplenomegaly has also been described [48].

Respiratory complications become more common in all children with MPS as the disease progresses [49]. Additionally, it has been proven that pneumonia is the cause of death in more than half of the children with MPS IIIA [50,51].

### 5.6. Skeletal System

Skeletal system problems do not occur in all patients with MPS. In some types, these symptoms are limited to short stature, while in others, they manifest as severe deformities of almost the entire skeleton. These symptoms are most visible and frequently diagnosed in patients with MPS IVA [52].

Patients with MPSPS also exhibit certain skeletal system disorders, one of the typical features being claw-shaped hands and wider metacarpal bones [16]. Similar disorders have been described in patients with severe forms of MPS I, II, and VII, which can eventually lead to a loss of hand function [3]. Additionally, MPSPS patients have been observed to have limited joint mobility and contractures, multiple deformities in bones other than the hands, and progressive skeletal dysplasia (dysostosis multiplex) [13,16,27]. Identical symptoms have been described not only in severe types of MPS I, II, and VII but also in MPS VI, where patients exhibit skeletal dysplasia, including short stature, dysostosis multiplex, and degenerative joint disease, while in MPS X patients, disproportionate short-trunk short stature and genu valgus were noted [53,54]. Patients with MPSPS also show joint inflammation and stiffness, as well as progressively worsening swelling of the lower limbs [15,27]. One patient also exhibited deformed femoral heads, excessive synovial membrane accumulation, and joint effusions [15]. Consequently, a gradual loss of motor skills was observed [15]. In some cases, vertebral flattening and metatarsal bone widening have also been documented [14].

The most significant skeletal system problems were observed in patients with MPS IVA. These symptoms include vertebral flattening, severe dysplasia of long bones (especially femoral epiphyses), joint instability, knee valgus, and resultant gait abnormalities [52]. However, a beta-galactosidase deficiency diagnosed as MPS IVB causes a considerably milder phenotype compared to MPS IVA [55]. It was reported that patients with MPS IVB exhibit limb dysplasia, but growth disturbances are less pronounced. Additionally, in these patients, enzymatic activity may be sufficient to prevent the development of peripheral nervous system symptoms observed in MPS IVA patients [56]. MPS IVA can be divided into three subtypes based on clinical presentation, depending on symptom severity. The severe form is characterized by systemic skeletal dysplasia identified at birth, while the mild form features slight skeletal disturbances and is usually diagnosed in adulthood. Children with the severe form often do not survive beyond the second or third decade of life, mainly due to cervical instability and impaired lung function [57,58,59].

The symptoms described in patients with MPS IX are usually mild, making proper diagnosis and appropriate treatment extremely difficult. The first diagnosed case of this type involved a 14-year-old girl who exhibited short stature, occasional soft tissue swelling, and periarticular soft tissue masses, while joint mobility was normal [60]. The other three patients diagnosed with MPS IX were siblings aged 11–21 years. One of them, at age 4 years, showed only hip or knee pain resulting from swelling. Further studies revealed synovitis and joint involvement [61].

### 5.7. Kidneys

Most of the symptoms observed in patients with MPSPS overlap with those in patients with other MPS types/subtypes. The organ where changes were observed only in MPSPS is the kidneys.

The term ‘plus’ refers to symptoms not found in other MPS types/subtypes. These additional symptoms include, in particular, kidney failure and hematopoietic system disorders. All patients with MPSPS are described as having nephrotic syndrome: enlarged kidneys and significant proteinuria [13,14,15,16,17,28], accompanied by hypoalbuminemia, creatinemia, elevated blood creatinine levels, and calcium deficiencies [16,28]. Most patients also show higher blood uric acid levels compared to healthy individuals [16,28]. All these data indicate kidney failure in MPSPS patients.

Histopathological studies were also conducted on some MPSPS patients. Some analyses even showed complete destruction of glomerular structures [28]. In milder cases, periglomerular fibrosis, interstitial inflammation, and the presence of foam cells in podocytes were observed [16]. Foam cells were also observed in MPS I and MPS VII; however, in the liver and spleen rather than the kidney. Some MPSPS patients also exhibit renal tubular diseases (tubulopathies) [17].

### 5.8. Other Symptoms

In MPSPS, several symptoms related to other systems/organs besides those mentioned above are also observed. Some patients have been noted to exhibit subclinical hypothyroidism [15,17]. Children with MPSPS have also been observed to have peripheral and retrocochlear hearing impairment [16,17]. Significant hearing impairment is also present in other MPS types/subtypes, which may be caused by middle ear infections, ossicular chain deformities, inner ear abnormalities, or auditory nerve impairment [62,63].

The summary of the main findings in MPSPS, in comparison to other MPS types, is presented in Table 1.

## 6. Similarities of MPSPS to Other Lysosomal Storage Diseases

Patients with MPSPS exhibited excessive accumulation of sphingolipids, β-D-galactosylsphingosine (psychosine), and the deacylated form of galactosylceramide, which also excessively accumulates in another LSD, Krabbe disease [64]. Psychosine excess is involved in the loss of oligodendrocytes and myelin damage, leading to severe disturbances in the white matter of the brain and peripheral nerves. Indeed, such disturbances have been described in patients with mutations in the *VPS33A* gene [13].

Fabry disease (FD) is a lysosomal storage disorder caused by deficiencies of the lysosomal enzyme α-galactosidase A, leading to the accumulation of globotriaosylceramide (Gb3) [65]. One of the main causes of death in FD patients is end-stage renal failure. The exact mechanism of kidney damage is not yet known, but it is hypothesized that it is caused by nephrotic syndrome resulting from the accumulation of Gb3 [28,66]. A very similar pattern of disorders has been observed in almost all patients with MPSPS [67,68,69] and in patients with ML III [28].

A mutation in the VPS33B gene leads to ARC syndrome (arthrogryposis–renal dysfunction–cholestasis), which includes renal tubular dysfunction, thrombocytopenia, and cholestasis [70]. The same mutation causes the ARKID syndrome (autosomal recessive keratoderma–ichthyosis–deafness), where patients also exhibit platelet dysfunction, ichthyosis, hearing loss, and osteopenia [71].

Symptoms caused by mutations in genes of other members of the HOPS and CORVET complexes differ phenotypically from those in MPSPS patients. A homozygous mutation in the VPS16 gene leads to primary dystonia during adolescence [72]. Loss of VPS18 functionality in mice leads to severe neurodegeneration and neuron-migration issues [73]. Joint contractures, delayed psychomotor development, and craniofacial deformities have been described in patients with mutations in the VPS8 gene [74]. Leukodystrophy caused by mutations in the VPS11 gene is associated with seizures, tetraplegia, or hearing loss [75,76,77]. However, this phenotype differs from that described in MPSPS patients. Leukodystrophy patients have been reported to exhibit increased urinary excretion of glycosphingolipids and elevated lysosomal storage. It was proposed that a defect in one of the HOPS complex components leads to substrate accumulation in lysosomes (including GAGs) but not to decreased lysosomal enzyme activity [13].

The p.Asp251Glu mutation in the *VPS33A* gene has been used in a mouse model for Hermansky–Pudlak syndrome [78,79]. Affected mice exhibit, among other things, severe behavioral disorders and degeneration of Purkinje cells [80].

Finally, it is worth noting that that similarities occur between some symptoms of MPSPS patients and those described in a recently discovered MPS X-type (deficiency in arylsulfatase K (ARSK) [81,82,83,84,85]. However, little similarity can be detected between MPSPS patients and those suffering from another recently recognized disease, MPS IIIE, though one should take into consideration the fact that there are serious doubts whether the latter disorder, caused by dysfunction of arylsulfatase G (ARSG), should be classified as MPS in humans or not [86].

## 7. Potential Therapies for MPSPS

Currently, there is no therapy for MPSPS. Regarding different MPS types, various therapeutical options were proposed, some of them being used in clinical practice [87]. Enzyme-replacement therapy (ERT) is available for clinical use for MPS I, MPS II, MPS IVA, MPS VI, and MPS VII. This therapy is based on the administration of recombinant human enzyme (which is lacking in the cells of patients) that is specifically modified to be efficiently recognized by the cellular receptor (usually the mannose-6-phosphate receptor), allowing the direction of the therapeutic protein to lysosomes [88]. Therefore, the recombinant enzyme can replace the function of the deficient one in the lysosomes of the patient’s cells. However, in MPSPS, the dysfunction concerns a non-lysosomal protein. Therefore, ERT cannot be effective, at least as long as a method for the efficient delivery of cytosolic proteins is lacking. Hematopoietic stem cell transplantation (HSCT) is another therapeutic option already in use for some MPS types [89]. Nevertheless, for reasons similar to those described for ERT, HSCT seems to be unlikely to work effectively in MPSPS. Gene therapy is an obvious potential cure for genetic diseases, and intensive work has been conducted to develop and introduce this kind of therapy to treat MPS patients [90]. However, no report has been published to date on the development of gene therapy targeting the *VPS33A* gene. Thus, even if such an effort is started, the putative introduction of gene therapy for MPSPS should take at least several years, making this kind of treatment a matter of the future rather than a promise for rapidly solving the medical problem.

Because of the problems mentioned above, it is perhaps more likely to develop an alternative therapy for MPSPS rather than focusing on ERT, HSCT, or gene therapy. One possible option would be to stimulate the autophagy process. Since the major cellular problem in this disease is extra-lysosomal accumulation of GAGs, with possible secondary storage of other compounds, activation of a process leading to the elimination of the storage material could be beneficial for patients. In fact, a potential use of autophagy stimulators has been proposed as a putative therapeutic approach in the treatment of MPS [91]. Such a strategy appeared recently to be effective in the case of cellular and animal models of another genetic disease caused by the accumulation of macromolecules, Huntington’s disease [92]. There are various potential autophagy stimulators that could be employed in long-term use (which is necessary in the case of genetic diseases), including already-tested genistein [92] or ambroxol [93], a drug previously approved for the treatment of other conditions. Obviously, extensive studies are required to test if the hypothesis about the efficacy of autophagy stimulation in the treatment of MPSPS is true.

## 8. Concluding Remarks

MPS are defined as diseases in which GAG is stored due to dysfunctions of lysosomal enzymes involved in the degradation of these compounds. MPSPS is a specific disease where accumulation of GAGs occurs despite normal activities of all lysosomal hydrolases and transferases involved in their decay. Thus, MPSPS does not fulfill the strict definition of MPS while being a disease revealing the common primary metabolic defect—inefficient removal of excess GAG. Despite genetic and biochemical differences, MPSPS and various MPS types share many common symptoms. On the other hand, some of them are specific to MPSPS, reflecting the etymology of this disease. Here, we propose a possible mechanism for GAG accumulation in MPSPS due to impaired vesicle trafficking and resultant improper localization of these complex carbohydrates, precluding their efficient degradation by lysosomal enzymes (Figure 1). If this hypothesis is true, it might explain the pathomechanism of MPSPS in more detail. Still, the question remains whether MPSPS is a type of MPS or should classified as a separate metabolic disease. The answer actually depends on how we define MPS. If both conditions (GAG storage and lysosomal enzyme dysfunction) must be met to call a disease MPS, then MPSPS is a separate disorder. If, however, we agree that the crucial feature of MPS is GAG storage and that the accumulation of GAG(s) is enough to include a disorder into the group of MPS, then MPSPS should be treated as another MPS type.

## Figures and Tables

**Figure 1 ijms-25-09570-f001:**
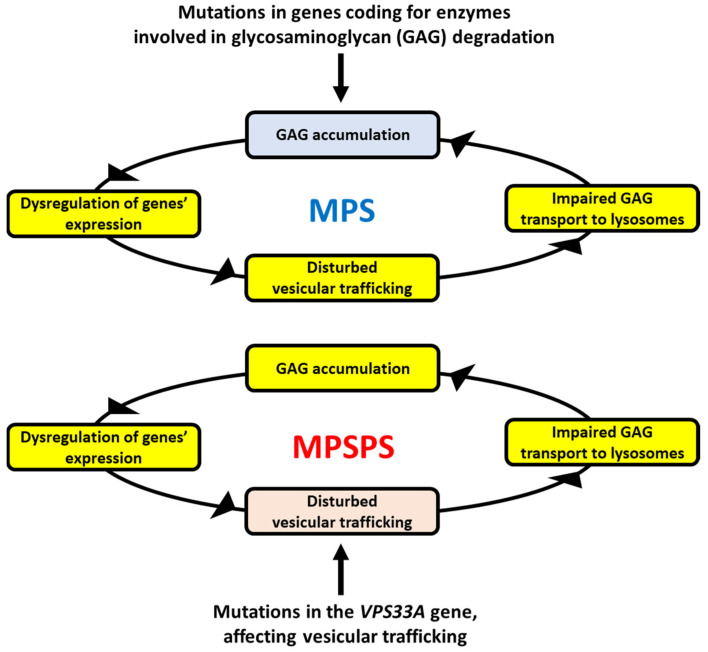
The hypothesis of the mechanisms of GAG accumulation in MPSPS. In classical MPS types (upper panel), GAG(s) accumulate(s) due to mutation(s) in one of genes coding for enzymes involved in degradation of this/these compound(s). GAG storage causes various secondary effects, among others, dysregulation of expression of many genes, resulting in pathological changes in different cellular processes. Disturbed vesicular trafficking is among them, leading to impaired transportation of GAGs into lysosomes. This makes GAG accumulation even more pronounced (as GAGs cannot be effectively degraded outside of lysosomes), which drives a spiral of above-described reactions within a positive feedback loop, enhancing the pathological processes. In MPSPS (lower panel), a similar spiral of reactions resulting in GAG accumulation occurs. However, it is initiated by impaired vesicular transport of GAGs (due to deficiency in VPS33A activity, which is required in this process) rather than dysfunction of an enzyme involved in GAG degradation. Nevertheless, the final effect (GAG accumulation) is similar in both classical MPS and MPSPS.

**Table 1 ijms-25-09570-t001:** The summary of the main findings of the MPSPS throughout life, with indication of the occurrence of similar symptoms in different MPS types.

Symptom(s)	Type of MPS	References
Characteristic facial features	All (including MPSPS)	[13,14,15,17,27]
Dysfunction of the liver and spleen	All (including MPSPS)	[11,13,14,15,27]
Prenatal ascites	MPS VI, MPSPS	[15,29]
Delayed psychomotor development with autistic symptoms	MPS III, MPS VII, MPSPS	[11,14,15,17,30,33,34]
Short stature	MPS IVA, MPS VI, MPS X, MPSPS	[17,35]
Delayed myelination, brain calcifications, retinal hypopigmentation, cerebellar abnormalities, and, in some cases, global brain atrophy	MPS I, MPS III, MPSPS	[13,16,17,37,38,39,40]
Mitral, aortic, and tricuspid valve regurgitation	MPS II, MPS X, MPSPS	[13,17,44,45]
Recurrent upper respiratory tract infections and pneumonia	MPS I, MPS II, MPS III, MPS IV, MPSPS	[11,13,15,17,46,47,48,50,51]
Claw-shaped hands and wider metacarpal bones	MPS I, MPS II, MPS VII, MPSPS	[3,16]
Dysostosis multiplex	MPS I, MPS II, MPS VI, MPS VII, MPS X, MPSPS	[13,16,27,52,53,54,55]
Enlarged kidneys, significant proteinuria	MPSPS	[13,14,15,16,17,28]

## Data Availability

No new data were created in this work.

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
