# Peer review of "Mucopolysaccharidosis-Plus Syndrome: Is This a Type of Mucopolysaccharidosis or a Separate Kind of Metabolic Disease?"

_ijms, 2024, doi:10.3390/ijms25179570_

Round 1

Reviewer 1 Report

Comments and Suggestions for Authors

A very interesting manuscript of great relevance, which will contribute to the "debate" of the classification and knowledge of the MPSPS.

I have two specific suggestions:

1. I believe that elaborating a figure or table that summarizes the main findings of the MPSPS throughout life (all of the current item 5) would contribute to a quick understanding of the message that is to be transmitted.

2. Nowhere in the manuscript is there any mention or comparison with MPS type 10, taking into account that this MPS is of recent description, I consider it appropriate to include it in the global comparison.

Aditionally, when the authors explain their hypothesis of the accumulation of GAGs in the MPSPS. I consider it appropriate to make an additional comment linking it with the differences in the metabolism of each of the GAGs, to give an integrated concept of the possible damage that occurs in the MPSPS

Author Response

COMMENT 1: I believe that elaborating a figure or table that summarizes the main findings of the MPSPS throughout life (all of the current item 5) would contribute to a quick understanding of the message that is to be transmitted.

RESPONSE 1: As recommended by the reviewer, Table 1 has been prepared which summarizes findings of the MPSPS, in comparison to similar symptoms occurring in other MPS types.

COMMENT 2: Nowhere in the manuscript is there any mention or comparison with MPS type 10, taking into account that this MPS is of recent description, I consider it appropriate to include it in the global comparison.

RESPONSE 2: We thank the reviewer for this comment. According to the request, comparison of MPSPS to MPS X has been incorporated into the text and Table 1. Moreover, a short note about this (and about a lack of similarities between MPSPS and another recently discovered disease, MPS IIIE) has been added to the text (lines 452-458).

COMMENT 3: Aditionally, when the authors explain their hypothesis of the accumulation of GAGs in the MPSPS. I consider it appropriate to make an additional comment linking it with the differences in the metabolism of each of the GAGs, to give an integrated concept of the possible damage that occurs in the MPSPS.

COMMENTS 3: As recommended by the reviewer, the differences in the metabolism of various GAGs are discussed, with special emphasis on the differences in their transportation, as all of them are degraded by lysosomal hydrolases, thus differences in activities between these enzymes in MPSPS are unlikely. This discussion in presented in lines 180-201 of the revised manuscript.

Reviewer 2 Report

Comments and Suggestions for Authors

This is an excellent review of MPSPS. While lysosomal enzyme activity in MPSPS is normal, the patients show abnormally high levels of GAG in the urine and display a combination of physical manifestations seen in different forms of MPS, in addition to kidney function failure and blood cells abnormalities. The disease appears to be caused by mutations in the VPS33A protein, leading to disturbances in vesicular trafficking. 

I would suggest to add information regarding the occurrence of corneal clouding (observed in some forms of MPS) and foam cells. The authors describe foam cells in the kidneys of some patients. However, foam cells are often observed in liver and spleen on MPS patients, caused by the expansion of lysosomes. It will be interesting to know whether these foam cells also appear without the apparent involvement of lysosome degradation in MPSPS patients. Finally, a short discussion of potential therapies could be included. Has protein replacement been considered, or gene therapy?

Author Response

COMMENT 1: I would suggest to add information regarding the occurrence of corneal clouding (observed in some forms of MPS) and foam cells. The authors describe foam cells in the kidneys of some patients. However, foam cells are often observed in liver and spleen on MPS patients, caused by the expansion of lysosomes. It will be interesting to know whether these foam cells also appear without the apparent involvement of lysosome degradation in MPSPS patients.

RESPONSE 1: As suggested by the reviewer, the presence of foam cells in the liver and spleen of MPS I and MPS VII is indicated in the revised manuscript (lines 403-404). Unfortunately, it is not known whether foam cells occur also in these organs of MPSPS. Although corneal clouding is characteristic for some MPS types, it was not described to date in MPSPS, thus, we did not mention it in this paper.

COMMENT 2: Finally, a short discussion of potential therapies could be included. Has protein replacement been considered, or gene therapy?

RESPONSE 2: As requested by the reviewer, a discussion on potential therapies for MPSPS has been included. It is presented in a new chapter 7 (entitled “Potential therapies for MPSPS”), in lines 459-492 of the revised manuscript.